# South Atlantic meridional transports from NEMO-based simulations and reanalyses

Davi Mignac<sup>1,2</sup>, David Ferreira<sup>2</sup>, and Keith Haines<sup>2</sup>

<sup>1</sup>Postgraduate Program in Atmosphere, Oceans and Climate, University of Reading, Reading, UK <sup>2</sup>Department of Meteorology, University of Reading, Reading, UK

Correspondence to: Davi Mignac (d.mignaccarneiro@pgr.reading.ac.uk)

# 10

5

# Abstract

The Meridional Heat Transport (MHT) of the South Atlantic plays a key role in the global heat budget: it is the only

- equatorward basin-scale ocean heat transport and it sets the northward direction of the global cross-equatorial transport. Its strength and variability however are not well known. The South Atlantic transports are evaluated for four state-of-the-art global Ocean Reanalyses (ORAs) and two Free-Running Models (FRMs) in the period 1997-2010. All products employ the Nucleus for European Modelling of the Oceans model, and the ORAs share very similar configurations. Very few previous works have looked at ocean circulation patterns in reanalysis products, but here we show that the ORA basin interior transports are
- 20 consistently improved by the assimilated in situ and satellite observations, relative to the FRMs, especially in the Argo period. The ORAs also exhibit systematically higher meridional transports than the FRMs, in closer agreement with observational estimates at 35°S and 11°S. However, the data assimilation impact on the meridional transports still greatly varies between the ORAs, leading to differences up to ~8 Sv and 0.4 PW in the South Atlantic Meridional Overturning Circulation and the MHTs, respectively. We narrow this down to large inter-product discrepancies in the Western Boundary Currents (WBCs) at both upper and deep levels explaining up to ~85% of the inter-product differences in MHT. We show that meridional velocity differences, rather than temperatures differences, in the WBCs drive ~83% of this MHT spread. These findings show that the present ocean observation network and data assimilation schemes can be used to consistently constrain the South Atlantic interior circulation, but not the overturning component which is dominated by the narrow western boundary currents. This will likely limit the effectiveness of ORA products for climate or decadal prediction studies.

# 1. Introduction

An important feature of present-day climate is that the heat transport in the Atlantic Ocean is northward in both hemispheres, rather than poleward as in the Indo-Pacific Ocean (Ganachaud and Wunsch, 2003) and in the atmosphere (Trenberth and

35 Caron, 2001). The South Atlantic acts as a communicator between the southern and northern oceans (Garzoli and Matano, 2011), through the Meridional Overturning Circulation (MOC) transporting warm water northward across the equator to compensate for the southward export of colder North Atlantic Deep Water (NADW).

The northward upper limb of the South Atlantic MOC (hereafter SAMOC) is a complex mixture of water masses originating from the Indian, Pacific and Southern oceans, which are blended together in the South Atlantic gyre circulations. The water

- mass redistribution in the South Atlantic and the interocean exchanges can significantly influence the long-term Atlantic MOC (hereafter AMOC) variability (Garzoli and Matano, 2011), particularly on decadal time scales through the heat and salt export by the Agulhas leakage (Weijer et al., 2002; Sebille et al., 2011). The SAMOC salt fluxes at 35°S have also been suggested to reflect the MOC stability in climate models (Drijfhout et al., 2011; Hawkins et al., 2011). In the case where the SAMOC imports salt into the Atlantic basin, a weakening of the AMOC would be followed by a further freshening of the basin, a positive feedback possibly leading to the collapse of the AMOC.
- Marshall et al. (2013) argue that the northward ocean heat transport across the equator sets the mean position of the Inter-tropical Convergence Zone in the northern hemisphere. Since the South Atlantic is the only major ocean basin that transports heat equatorward, quantifying and understanding the SAMOC should help to explain the inter-hemispheric heat exchanges and improve interannual-to-decadal climate simulations, as also recently reinforced by Lopez et al. (2016). For this reason, a
  SAMOC observing system has already been initiated in 2002 with quarterly high-density eXpendable BathyTermograph (XBT) lines at 35°S (Garzoli and Baringer, 2007), and recently with the development of the South Atlantic MOC Basin-wide Array (SAMBA; Ansorge et al., 2014), analogous to the RAPID array in the North Atlantic (Cunningham et al., 2007). However, the lack of long term measurements in the area still limits our understanding of the South Atlantic state and its variability, reflecting the large disagreement between observational and model studies (Garzoli et al., 2013; Dong et al., 2014;
- Dong et al., 2015; Majumder et al., 2016; Stepanov et al., 2016). In this context, Ocean Reanalyses (ORAs; Balmaseda et al., 2015) could be useful tools to monitor the ocean circulation and change indicators (Masina et al., 2015; Palmer et al., 2015).

The ORAs employ Ocean General Circulation Models (OGCM) and Data Assimilation (DA) schemes to synthetize a diverse network of available ocean observations in order to arrive at a consistent estimate of the historical ocean state. In such products, atmospheric forcing combined with DA are used to dynamically extrapolate the observational information to regions without

observations, which gives the ORAs the potential to provide complete, time-evolving descriptions of the ocean state and its circulation.

In the South Atlantic, ORA diagnostics have been put together with three-dimensional velocity fields constructed from Argo and Sea Surface Height (SSH) observations to study the SAMOC variability and its relation with the Meridional Heat Transports (MHT) between 35°S and 20°S (Majumder et al., 2016). Although both observations and ORAs show strong correlations between the SAMOC and MHT, Majumder et al. (2016) also found significant discrepancies in the transport magnitudes between the ORAs as well as between the ORAs and the observations. Their result reveals the need for further assessment of the skills and uncertainties of the ORAs in the South Atlantic, such as comparing them with Free-Running

Models (FRMs) and evaluating their SAMOC contributions across the eastern, interior, and western boundary regions shown

in Fig. 1.

The next generation of operational climate prediction systems will implement eddy-permitting ocean models, and it is expected that ORAs will provide improved initial conditions for such climate prediction models. The comparison between ORAs and FRMs is a critical step in assessing the feasibility of initialising the ocean transports which are not directly observed. Such intercomparisons therefore can give valuable insights about how the transports are affected by DA (e.g. Zuo et al., 2011; Karspeck et al., 2015). To address these issues, we use state-of-the-art ORAs at eddy-permitting resolution with two FRMs at eddy-permitting and eddy-resolving resolutions to study the meridional transports in the South Atlantic between 35°S and the equator. Focusing on the meridional volume and heat transports, we first identify similarities and differences between products. Going further than Majumder et al. (2016), we also narrow down these transport differences in an attempt to understand the potential impact (and limitations) of the DA schemes in improving the ORA states in the South Atlantic.

The paper is organised as follows. In Section 2 a brief overview of the dataset configurations is presented. Sections 3.1 and

3.2 show the results of the time mean transports and the contributions of the temperature (T) and meridional velocities (v) to the spread in the heat transports, respectively. Section 3.3 evaluates the western boundary role in the South Atlantic large-scale

transport discrepancies between the products. Section 3.4 ends the results section with the time variability of the transports. Section 4 contains the discussion and conclusions.

## 85 2. The dataset

In this study, we use outputs from two FRMs and four ORAs, each with a global domain. All the products are configured with the Nucleus for European Modelling of the Oceans (NEMO; Madec, 2008) model, coupled to the Louvain la Neuve sea-ice model version 2 (LIM2; Timmermann et al., 2005). The former is a state-of-the-art primitive equation z-level model employing both hydrostatic and Boussinesq approximations, whereas the latter is a dynamic-thermodynamic sea-ice model specifically designed for climate studies. For this dataset, NEMO is configured with a partial cell topography (Adcroft et al., 1997), and a quasi-isotropic tripolar ORCA grid (Madec and Imbard, 1996). Sub-sections listing the main characteristics of the FRMs and ORAs are presented below. Tab.1 compares the main configurations of each product.

## 2.1 Free-running models

- The standard configurations of the FRMs at 1/4° and 1/12° horizontal resolution used in this study have been setup within the DRAKKAR consortium (e.g. Barnier et al., 2006; Penduff et al., 2007, 2010; Treguier et al., 2014; Marzocchi et al., 2015). The FRM at 1/4° horizontal resolution is referred to here as ORCA025 and has 46 z-levels, with thickness ranging from 6 m at the surface to 250 m at the ocean bottom. ORCA025 is forced by the ERA-Interim atmospheric reanalysis product (Simmons et al., 2007) from the European Centre for Medium-Range Weather Forecasts (ECMWF). The ERA-Interim reanalysis provides Winds (W) at 10 m, Surface Air Temperature (SAT) and Surface Air Humidity (SAH) at 2 m, daily Radiative Fluxes (RF) and Precipitation (P) fields, which are used to compute 6-hourly turbulent air/sea fluxes using the Large and Yeager (2004, 2009) bulk formulae. The integration of this ORCA025 setup was conducted at the University of Reading and is described in Haines et al. (2012) and Stepanov and Haines (2014) as the free control run associated with reanalysis UR025.3, and its initial condition is derived from a previous 1/4° run with hydrographic data assimilation (Smith and Haines, 2009). A moderate relaxation of Sea Surface Salinity (SSS) is applied towards Levitus (1998) with a time scale of approximately 180 days.
  - 4

The FRM at 1/12° horizontal resolution (ORCA0083) has 75 z-levels. Its vertical grid is refined at the surface (1 m for the first level), smoothly increasing to a maximum thickness of 200 m at the bottom. The integration of ORCA0083 was performed by the Marine Systems Modelling group at the National Oceanography Centre, Southampton, and is described in Marzocchi et al.

(2015). The DRAKKAR Surface Forcing Set 4.1 (DFS4.1) or Set 5.1 (DFS5.1) is employed depending on the time period as shown by Tab.1. As detailed in Brodeau et al. (2010), DFS combines elements from two sources: (i) the Coordinated Ocean Research Experiments (CORE) forcing dataset, from which daily RF and monthly P are extracted; and (ii) ECMWF products from which W, SAT and SAH fields are taken. As in ORCA025, 6-hourly momentum and heat turbulent fluxes are computed in ORCA0083 following Large and Yeager (2004, 2009). ORCA0083 is initialised from Levitus (1998) climatology and applies the same SSS restoring term as in ORCA025. Both ORCA0083 and ORCA025 employ a free-slip (no-stress) configuration for the lateral momentum boundary conditions.

#### 2.2 Ocean Reanalyses

The MyOcean global ocean reanalysis activity provided a series of global ORAs at eddy-permitting resolution (1/4°)
constrained by assimilation of observations and covering the "altimetric era" (i.e. period starting with the launch of TOPEX POSEIDON and ERS-1 satellites at the end of 1992). Four of these ORAs are considered in this work, namely: (i) The Ocean Reanalysis Pilot 5 (ORAP5; Zuo et al., 2015) from ECMWF; (ii) The Global Ocean Reanalysis System version 5 (CGLORSV5; Storto and Masina, 2016) from the Centro Euro-Mediterraneo sui Cambiamenti Climatici (CMCC); (iii) The University of Reading Reanalysis Version 4 (UR025.4, Valdivieso et al., 2014); and (iv) The Global Ocean Reanalysis and Simulation Version 4 (GLORYS2V4; CMEMS, 2017) from Mercator Ocean. These ORAs employ different state-of-the-art ocean DA systems, which assimilate, in distinct ways, reprocessed observations of Sea Level Anomaly (SLA), Sea Surface Temperature (SST), in situ T and Salinity (S) profiles, and Sea Ice Concentration (SIC). The main references of the ORA DA schemes and their assimilated observations can be found in Tab.1 The vertical discretisation of GLORYS2V4, ORAP5 and UR025.4 follows exactly the same configuration as in ORCA0083 with 75 z-levels. CGLORSV5 has 50 z-levels in a similar

configuration to ORCA025.

All the ORAs are forced with the ERA-Interim atmospheric reanalysis product from ECMWF. The turbulent air-sea fluxes were calculated using the same methodology as in the FRMs, but their input into NEMO varies between 3 and 6-hour sampling depending on the product (see Tab. 1). In GLORYS2V4, large-scale corrections of the atmospheric forcings are also applied (Garric and Verbprugge, 2010), whereas in ORAP5 the impact of surface wave forcing on the ocean mixing and circulation is included (Janssen et al., 2013).

The relaxation strategies differ between the ORAs. In ORAP5 and CGLORSV5, the SST data in Tab. 1 are used to correct the turbulent heat fluxes, with a restoring term of -200 W m<sup>-2</sup> K<sup>-1</sup>. Their SSSs are also relaxed towards the World Ocean Atlas 2009 (WOA09; Locarnini et al., 2010) for ORAP5, and towards the UK Met Office ENhAnced ocean data assimilation and ClimaTe prediction (ENACT/ENSEMBLES) EN4 dataset (Good et al., 2013) for CGLORSV5, with time scales of approximately 300 days. No global SST and SSS restoring strategies have been implemented in UR025.4 and GLORYS2V4, and the only surface restoring mechanism is through the increments introduced by data assimilation itself. As also seen in Tab.1, the initialisation and spin-up differ between the ORAs. On lateral boundaries, UR025.4 and ORAP5 adopt a free-slip configuration whereas CGLORSV5 and GLORYS2V4 employ a partial-slip condition. In the latter, the constant of proportionality (α) between the tangential stress and the tangential velocity is defined as 0.5 for both products. More specific details comparing these NEMO-based ORAs can be found in Masina et al. (2015).

In this work, monthly averages of each product are used. The use of monthly means mitigates possible jumps introduced by incremental assimilation over a time window of several days. In order to avoid any dynamical spin-up in the early years of the simulation for products starting in the late eighties or early nineties (e.g. UR025.4 and GLORYS2V4), and because UR025.4 ends in 2010, a common time period from 1997 to 2010 is chosen. Despite the fact that subsurface ocean observations are

150 scarcer before the 2000s (i.e. prior to the full deployment of Argo floats), the total meridional transports for the periods 1997-2010 and 2000-2010 do not differ significantly.

# 2.3 Observational estimates and surface heat flux products

135

The large-scale transports are compared to the 34 high-density XBT-based estimates (XBT-AX18) in the Southern Atlantic from 2002 to 2013, with transport estimates at 35°S and 30°S given by Majumder et al. (2016). Recent observational studies are also used for comparison, which employ different methodologies to calculate the SAMOC and MHT between 35°S and 20°S, as follows: (i) an Argo climatology (Dong et al., 2014), (ii) altimetry synthetic profiles based on the correlation of the AVISO SLA and isotherm depths (Dong et al., 2015), and (iii) dynamic height fields from Argo and AVISO SSH (Majumder et al., 2016) are used together with wind fields to estimate the total transports. The MHT based on integrating the Liu et al. (2015) surface heat flux product southward of 80°N is also computed for the 1997-2010 period. This product uses top of atmosphere net radiation flux from CERES modified by the ERA-Interim atmospheric transports. The North Brazil Current (NBC) transports from 2000 to 2004 (Schott et al., 2005) and from 2013 to 2014 (Hummels et al., 2015) are also included for

comparison. These NBC estimates are based on high-frequency velocity measurements from a moored western boundary array section located at 11°S. Finally, WOA13 temperatures (Locarnini et al., 2013) from 1995 to 2012 are also compared with the
 temperatures from the ORAs and FRMs.

Of the observational estimates above, the XBT-AX18 line is not independent as it is included in the EN3 and EN4 datasets which are assimilated by the ORAs (see Tab.1). Although WOA13 is not directly assimilated by the ORAs, it uses the same observational information as EN3 and EN4, and so it also cannot be treated as completely independent.

# 170 **3. Results**

## **3.1 Time-mean transports**

Figure 2a shows the time mean AMOC strength for each product, defined as the maximum (\u03c6<sub>max</sub>) of the AMOC stream function at each latitude in the Atlantic basin. The ensemble spreads of \u03c6<sub>max</sub> for all products (ENS-ALL hereafter) and for only the ORAs (ENS-ORA hereafter) are shown in Fig. 2b. The discrepancies in AMOC strength between the ORAs are largest
in the South Atlantic, reaching the maximum spread of 3.5 Sv (ENS-ALL) and 3 Sv (ENS-ORA) in the area between 20°S and the equator. The two FRMs are similar to each other, both with relatively low AMOC across the basin. The assimilation of observations in the reanalyses appears to increase the AMOC strength at all latitudes. In the North Atlantic, especially in the subpolar gyre north of ~35°N, the ORA AMOCs are consistently 3-4 Sv higher than in the FRMs. However, the increase of the ORA AMOCs is less consistent south of 35°N, especially in the South Atlantic where the differences in the SAMOC

transports can reach up to ~8 Sv between GLORYS2V4 and ORAP5. The latter is the ORA that has the lowest transports in the South Atlantic, closest to the FRMs.

Comparison with observational estimates at 35°S (Figs. 3a-b) suggests that both the SAMOC strength and MHT of the ORAs are more realistic than those of the FRMs. However, even the highest MHTs of UR025.4 and GLORYS2V4 are almost 0.1 PW lower than the lowest observational estimate from Dong et al. (2015). The MHT underestimation of the FRMs and ORAs

- relative to the observations at 35°S has already been reported by several authors (e.g. Dong et al., 2011a; Dong et al., 2011b; Perez et al., 2011; Sitz et al., 2015; Majumder et al., 2016; Stepanov et al., 2016). The black bars in Figs. 3a-b show monthly variability in the ORAs, but quarterly (XBT-AX18), monthly (Dong et al., 2014), weekly (Dong et al., 2015) or daily (Majumder et al., 2016) time scale variability in the observations. These clearly overlap each other although they cannot be regarded as uncertainties in the means. Despite their lower mean transports, the temporal variability of the FRMs is similar to
- that of the ORAs at 35°S, around  $\pm 0.3$  PW and 3.0 Sv.

As in the SAMOC strength (Fig. 2), the inter-product spread in MHT gets larger towards the equator, with differences up to 0.4 PW between GLORYS2V4 and ORAP5 (Fig. 3c). The Liu et al. (2015) surface flux based product suggests higher heat transports in good agreement with UR025.4 and GLORYS2V4 across the South Atlantic basin, although the surface integration method accumulates errors from all higher latitudes. Liu et al. (2015) estimates also reasonably agree with the XBT-AX18 and other South Atlantic observational studies at 35°S and 30°S. However, the observational estimates diverge north of 30°S, with

195 other South Atlantic observational studies at 35°S and 30°S. However, the observational estimates diverge north of 30°S, with the transports from Dong et al. (2015) and Majumder et al. (2016) differing by ~0.7 PW at 20°S. These discrepancies underscore the uncertainties in observed transports through the South Atlantic.

Figures 4a-f show maps of the east-west accumulated volume transports from the surface down to the depth of  $\psi_{max}$  (typically ~1000 m) for each latitude, defined hereafter as  $z_{max}$ . These contours can be regarded as streamlines of the upper ocean gyre circulations. The northern boundary of the subtropical gyre (dashed contour of zero transport), near 20°S and 15°S, agrees well between products, with only GLORYS2V4 extending slightly further north. The subtropical gyre to the south is only partially shown but the strength of this gyre is quite consistent between the ORAs and ORCA0083, and significantly stronger than in ORCA025. The large-scale circulation equatorward of 15°S is dominated by a southward flow increasing westwards until the strong northward NBC flow is reached in a very narrow western boundary area. The ORA southward flow in the basin interior

ranges between -14 and -18 Sv. For consistency with the overturning strength  $\psi_{max}$  (represented in Figs. 4a-f by the westernmost accumulated transports), the NBC region typically reaches ~36 Sv of northward flow. This agrees with other studies of the role of the NBC in the AMOC upper branch crossing the equatorial Atlantic (e.g. Rabe et al., 2008; Sebille et al., 2011; Rühs et al., 2015).

Figure 4g shows the southward maximum of the east-west accumulated transports between 15°S and the equator. The generally good agreement of this interior component of the circulation between the ORAs is in striking contrast with their  $\psi_{max}$  (Fig. 2). Indeed the ENS-ORA spread of the interior flow (~1 Sv) is about three time less than the spread in  $\psi_{max}$  for the same latitude range. The ORA southward transports differ from the FRMs, with two peaks of southward transport between 10°S and the equator where the FRMs only have one. The zonal currents, which can be inferred in Fig. 4, reveal consistent changes in the equatorial current system between the ORAs and the FRMs. The central branch of the South Equatorial Current (cSEC),

described in the top 500 m tropical circulation schematics of Stramma and Schott (1999) and Talley (2011), is absent in the FRMs, but evident in the ORAs, also leading to stronger southward transports in Fig. 4g. Thus there is both qualitative and quantitative evidence that the DA in the ORAs is doing a good job in reproducing a consistent interior circulation for the tropical South Atlantic basin.

Despite evidence of the ORAs consistency in the interior circulation in the tropical South Atlantic as well as in the subtropical gyre further south, the overturning transport component  $\psi_{max}$ , associated with the very narrow NBC, is not as well constrained. Figure 5 shows transports of the NBC at 11°S, calculated between neutral density interfaces as in Hummels et al. (2015). Although DA brings the ORA NBC transports closer to the observations when compared to the FRMs, the spread is still large. The UR025.4 and GLORYS2V4 NBC transports have  $23.9 \pm 1.1$  Sv and  $25.0 \pm 1.3$  Sv, quite close to the Schott et al. (2005) and Hummels et al. (2015) observed NBC values of  $25.8 \pm 1.2$  Sv and  $26.8 \pm 1.8$  Sv, respectively. However, the weaker

transports in ORAP5 and CGLORSV5 mean that the ENS-ORA spread in the NBC transports is ~3 Sv, which is consistent with the ENS-ORA spread in the SAMOC strength (Fig. 2b). This suggests that, at least in this latitude range, the NBC strength alone can explain the large-scale transport discrepancies between the ORAs, which will be discussed in more detail in Sect. 3.3.

#### 230 3.2 T and $\nu$ contributions

In this section, the contributions from T and v variability for the heat transports are analysed, as well as the relationship between the MHT and the SAMOC upper limb. Figure 6 shows a meridional section of the zonal-mean temperatures from WOA13, together with zonal-time mean anomaly T from each product. Large anomalies in the FRMs can be seen, particularly in the tropics where the models may have limitations representing sharp vertical gradients in the tropical thermocline. In ORCA025, there is a large warm anomaly of up to 3°C in the upper 200 m of the tropical South Atlantic, whereas ORCA0083 has a weaker warm anomaly in the top 200 m, but a much more extensive cold anomaly of ~2°C in the ocean interior down to ~500 m. All the ORAs show much weaker anomalies (mostly <0.5°C), presumably due to the assimilation of SST and T/S profiles which are able to better constrain the T vertical structure. Below 1200 m the differences between the products and WOA13 are much smaller.

- Figure 7 evaluates the relative T and v contributions to the ENS-ALL MHT spread. We compare the original MHTs (Fig. 7a) with the MHTs based only on circulation differences (vT; Fig. 7c), and only on temperature differences (vT; Fig. 7e), where the overbar denotes the ENS-ALL mean. In order to identify locations where T and v contribute to different transports in ENS-ALL, ocean temperature transports per 0.25° of longitude (p-OTTs) from top to bottom are also calculated across the basin (Fig. 7b), with their p-vT (Fig. 7d) and p-vT (Fig. 7f) contributions. Note that the units in the maps of Figs. 7b,d,f are PWT
  (PetaWatt Temperature Transport; Talley, 2003; Macdonald and Baringer, 2013) per 0.25°. The spatial discretisation of the
  - MHT on a longitudinal 0.25° grid allows to present ORCA0083 on a comparable scale to that of the other models.

The strong similarity between Figs. 7a,b and Figs. 7c,d reveals that v rather than T differences drive the inter-product spread in the MHTs, both regionally and in the zonal integrals. The  $v\overline{T}$  component captures variations from ~0.2 PW to 1 PW (Fig. 7c), explaining ~83% of the total MHT spread which is mainly concentrated in the areas with largest mean transports, i.e. the narrow western boundary region (Fig. 7d). Even with relatively large T anomalies found in the FRMs (Fig. 6), the  $v\overline{T}$ component only differs by ~0.13 PW between the products across the basin (Fig. 7e), mainly due to temperature differences in ORCA025 and ORCA0083. However, a very narrow maximum of p- $v\overline{v}T$  (Fig. 7f) can also be seen right against the western boundary, especially in the NBC region around 11°S and near the Brazil-Malvinas Confluence at 35°S. This is interpreted as due to variations in boundary temperatures needed to geostrophically support the large differences in western boundary current velocities between the products. However, these temperature differences make little transport contribution. The detailed role of the western boundary for the inter-product transport discrepancies will be discussed again in Sect. 3.3.

The dominance of the circulation determining heat transports also extends to the time variability. The monthly correlation between  $\psi_{max}$  and MHT within all products is above 0.8 for most of the South Atlantic (Fig. 8). Dong et al. (2009) and Garzoli et al. (2013) estimated quarterly correlation values around 0.75 between circulation and heat transports at 35°S from the XBT-

- AX18 observations. Majumder et al. (2016) found that a 1 Sv change in the SAMOC strength corresponds to a change of 0.046 PW at 35°S and 0.056 PW at 20°S in the MHT. This agrees relatively well with the ENS-ORA which show a 1 Sv change in SAMOC strength corresponds to ~0.052 PW change between 35°S and 20°S. It is interesting to note that correlations abruptly fall from 0.85 to ~0.45 near the equator. The interior southward flow gradually increases in the tropical South Atlantic reaching similar magnitudes to ψ<sub>max</sub> between 5°S and the equator (Fig. 4g). In this region, the temperature differences between the NBC core and the southward basin interior circulation reach up to 5.5°C in the top 400 m, similar to the ΔT of ~6.5°C between
- NBC core and the southward basin interior circulation reach up to 5.5°C in the top 400 m, similar to the  $\Delta T$  of ~6.5°C between the SAMOC upper and lower limbs (not shown). Therefore it is likely that these large upper level tropical circulations explain why  $\psi_{max}$  does not dominate the MHT variability close to the equator, as also noted by Valdivieso et al. (2014).

#### 3.3 Western boundary contribution

Figure 9 shows the linear regression coefficient between the inter-product p-OTTs and their MHTs across the whole basin. The western boundary grid points in the tropical South Atlantic reach up to ~0.4 PWT per 0.25°, out of 1 PW across the whole basin, so that ~40% of the differences in the MHT can be explained by transports in a 0.25°-wide band (a single grid point in all models except ORCA0083), with values elsewhere in the basin interior very close to zero. This is consistent with Fig. 4 showing that the large-scale southward flow at upper levels does not differ much between products, while ψ<sub>max</sub> varies
considerably, mainly due to the narrow NBC. Weaker negative linear regression coefficients are found eastward of the NBC in Fig. 9, representing the influence of the southward Deep Western Boundary Current (DWBC), reflecting the sloping bathymetry and the broader current scale than the NBC. South of 25°S the p-OTT contributions to the total MHT are more distributed, with a noticeable contribution from the Agulhas leakage caused by the different intensity and positioning of the Agulhas rings between the products as they travel westward across the Cape basin. Figure 9 also shows a continuous and

dominant narrow band of positive regression coefficients all down the western boundary, including latitudes where the p-OTTs have a southward transport associated with the Brazil Current (BC), e.g. between 35°S and 25°S (see schematic of Fig. 1). This reveals that products with larger northward MHTs (e.g. CGLORSV5, UR025.4 and GLORYS2V4) must have weaker southward p-OTTs near the western boundary, i.e. a weaker BC, resulting in the positive MHT linear regressions. In the case of CGLORSV5, UR025.4 and GLORYS2V4, this is reinforced by a stronger northward subsurface transport of the Intermediate Western Boundary Current (IWBC) and North Brazil Undercurrent (NBUC), which feeds the NBC in the tropical South Atlantic (Fig. 10a and Fig. 10b). Based on Fig. 9, a region within 6° of the coast is selected to calculate the Tropical Water (TW), South Atlantic Central Water (SACW) and Antarctic Intermediate Water (AAIW) transports of the upper western boundary circulation, with their isopycnal limits defined as in Mémery et al. (2000) and Donners et al. (2005). For each latitude, any southward water mass transport is accounted for as the BC (Fig. 10a), whereas any northward transport contributes to the IWBC-NBUC-NBC system (Fig. 10b), allowing to represent the deepening of the poleward BC and the shallowing of the

equatorward IWBC-NBUC-NBC flows, as shown by Fig. 1 (Soutelino et al., 2013).

In GLORYS2V4 and UR025.4, the IWBC and NBUC transports are at least 5 Sv larger than in ORAP5 and the FRMs (Fig. 10b), and the former products then produce a stronger NBC in the tropical South Atlantic, consistent with the observational estimates at 11°S (Fig. 5). At each latitude the ORAs usually modify the upper western boundary circulation in the same

direction, increasing (decreasing) the transports of the northward (southward) currents compared to the FRMs, which leads to higher MHTs across the entire basin. However, the western boundary transport magnitudes are not properly constrained in the ORAs, as reinforced by Fig. 10c, with the ENS-ORA spread increasing as current strengths increase. The IWBC-NBUC-NBC spread particularly growths from ~1 to 3.5 Sv towards the north which is comparable to the SAMOC spread seen in Fig. 2b. There is much better agreement for the BC near 35°S between the ORAs (ORAP5 excepted), with spreads smaller compared

to the NBC.

In Fig. 11, the transports are schematically broken down into four boxes, the upper and lower western boundary region (within  $6^{\circ}$  of the coast), and the upper and lower ocean interior ( $z_{max}$  separates the upper and lower layers). Figure 11 summarises how the inter-product changes in the upper western boundary circulation correlate with the other three boxes (for the current systems involved see Fig. 1). In the tropical South Atlantic (Fig. 11a), the northward flows in the upper western boundary box

- in GLORYS2V4 and UR025.4 are ~10 Sv and 8.5 Sv larger than in ORCA025, respectively. These are mainly compensated by larger flows in the DWBC, by ~9 Sv and 8 Sv in GLORYS2V4 and UR025.4, respectively, relative to ORCA025. These large inter-product compensations confined to the western boundary extend to the subtropical region (Fig. 11b), where the ORAs with highest southward DWBC transports show highest northward transports in the western boundary upper limb. Similarly, Sitz et al. (2015) found that the strengthening in the SAMOC upper limb with increasing model resolution is mainly
- compensated by strengthening of the poleward transport in the deeper layers, mostly in the western part of the basin. This large compensation between the upper and lower western boundary circulation is evident within all products in Fig. 11a, with the deep western boundary typically compensating ~75% of its upper limb transports, which was also noted in observations (Schott et al. 2005; Hummels et al. 2015).

In contrast to their western boundary circulations, the ORAs show very similar upper interior flows across the South Atlantic,

consistently stronger than in the FRMs, regardless of direction (southward in Fig. 11a and northward in Fig. 11b). This consistency is retained even in the subtropical gyre (Fig. 11b), where the northward basin interior circulation can have larger magnitude than the upper western boundary currents to balance the DWBC. The deep interior box has negligible transports in the tropical South Atlantic, but significant southward transports further south, especially in the ORAs, suggesting that some portion of the NADW flows towards the interior of the basin in the subtropical South Atlantic (Garzoli et al., 2015).

#### 3.4 Temporal variability

Figures 12a-f show that the interannual variability in p-OTTs is larger in the ORAs and in the high resolution ORCA0083 than in ORCA025. The assimilation of observations in eddy-permitting models introduces variability that would otherwise only appear with higher resolution, as in ORCA0083. According to Masina et al. (2015), this higher variability in the ORAs is in better agreement with the Eddy Kinetic Energy estimates from the ocean surface current velocities (OSCAR) product than that of the FRMs. Although some of the ORAs have more transport variability than others throughout the basin, the western boundary variability remains a dominant feature, particularly northward of 25°S. In Fig. 12g, the interannual p-OTTs variances for each product are summed within 6° of the western boundary coast as a function of latitude and displayed as a percentage

of the total MHT variance. It shows that the western boundary controls ~70% of the interannual MHT variability in the tropical

- South Atlantic for almost all the products (UR025.4 excepted), but it is less dominant further south.
  - South of 25°S, the interannual variability of the transports is more spread, with contributions from the western boundary (near the Brazil-Malvinas confluence), and near the eastern boundary (due to the Agulhas leakage) with the largest values around 0.06 PWT per 0.25° in ORCA0083, UR025.4 and GLORYS2V4. The different levels of variability in the Agulhas leakage between ORCA025 and ORAs may be attributed to the impacts of the SLA assimilation (Backeberg et al., 2014). However, even between ORAs these Agulhas patterns differ, e.g. the weaker contributions in ORAP5 may be due to smoothing from the super-observation method applied to the altimeter data (Mogensen et al., 2012), as also noted by Masina et al. (2015).
- Figure 13a shows the monthly time series of both  $\psi_{max}$  and the maximum southward flow in the basin interior (as in Fig. 4g), as a spatial average from 15°S to the equator. There appears to be greater consistency in the ORA southward transports in the second half of the time series, which is not seen in  $\psi_{max}$ . In Fig. 13b, the time series of the ENS-ORA spread for both 340 components are also displayed. A running mean of 6 months was applied to smooth the ENS-ORA monthly variability. Even with large variations, particularly in the first years of the time series, the ENS-ORA spread for the upper southward flow is seen to reduce from ~3 Sv to 1 Sv in the later years. This may be explained by the initiation of the Argo program and the increased number of observations to constrain the southward interior flow in the ORAs. The southward interior transports in the ORA maps of Fig.4 from 2008 to 2010 are also more consistent than before 2002, as are their northward interior transports between 30°S and 15°S in the later years (not shown). However, the ENS-ORA spread in  $\psi_{max}$  remains nearly steady over 345 this period, although the assimilation does increase the NBC transports in the ORAs relative to the FRMs (Fig. 5).

# 4. Discussion and conclusions

In this work, the South Atlantic meridional transports between 35°S and the equator were evaluated for a global NEMO-based dataset of four ORAs and two FRMs with distinct spatial resolutions. The ORAs mainly differ by their initial conditions, their DA schemes and to some small extent by the observations assimilated, as they share very similar ocean model configurations and are all forced with the ERA-Interim atmospheric product (Tab. 1).

Some aspects of the circulation are well constrained by data assimilation. The ORA transports in the basin interior are consistently modified across the basin relative to the FRMs (Fig. 4 and Fig. 11), with improvements in the south equatorial

- currents, and with interior meridional transports converging as Argo data are introduced (Fig. 13). Zonally integrated temperature sections for the ORAs are also very similar to WOA13 (Fig. 6), whereas the FRMs have large anomalies. The relationship between the magnitudes of SAMOC and MHT in the ORAs is in good agreement with that inferred in observations (e.g. Garzoli et al., 2013; Majumder et al., 2016), and the SAMOC upper limb and MHT are also strongly correlated in time at most latitudes (Fig. 8).
- The DA does appear to systematically increase the ORA SAMOCs and MHTs with respect to the FRMs, bringing them closer to observational estimates at 35°S and western boundary measurements at 11°S (Fig. 3 and Fig. 5). The assimilation of Argo data, for example, leads to a significant intensification of the boundary currents relative to the pre-Argo period and to an improvement in the SAMOC structure at 35°S in comparison with XBT-AX18 estimates (see also Dong et al. (2011a)). Here, although the DA consistently changes the upper western boundary transports in the same direction (e.g. increasing the northward IWBC-NBUC-NBC and decreasing the southward BC), they do not consistently constrain the boundary current transport magnitudes. Large SAMOC and MHT discrepancies still remain between the ORAs. These discrepancies are mainly attributed to differences in the narrow South Atlantic western boundary currents found within a few degrees of the coast. For example, the NBC (15°S-equator) explains ~85% of the inter-product differences in the total MHTs, with compensating variations in the return flow (DWBC) also close to the coast. Since the overturning stream function ψ<sub>max</sub> is mainly associated
- with these boundary flows, it is not well constrained by the ORAs, particularly in the tropical South Atlantic.

Analysis of the heat transports also reveals that differences in transport rather than differences in temperature dominate the inter-product spread, even within the western boundary region. The temperature contribution to the inter-product spread in heat transport,  $\bar{v}T$ , is only ~17% of the total spread, but its signature is evident right against the western boundary where temperature differences are required to geostrophically support the velocity differences between products. The local response to small density changes on the western boundary slope was also found to largely determine the meridional transport variability in ocean models in the North Atlantic, as noted by Bingham and Hughes (2009), emphasising the large sensitivity of the currents with respect to local density gradients against the boundary.

It is noteworthy that the lateral boundary conditions in the ORAs and FRMs vary between free-slip ( $\alpha$ =0) and partial-slip ( $\alpha$ =0.5). However, there is no clear correspondence between the choice of lateral boundary conditions and the strength of the

western boundary transports, with free-slip products (e.g. UR025.4) having similar transports to partial-slip products (e.g. GLORYS2V4).

Two possible reasons for the ORA differences in the western boundary currents are: (i) the lack of near boundary observations, and/or (ii) the differences in DA error covariances when assimilating interior basin measurements lying near to the western boundary. Observation system simulation experiments (OSSEs) with AMOC trans-basin arrays have shown that the meridional flow strength can be sensitive to the number of hydrographic profiles near the boundaries in both North (e.g. Hirschi et al., 2003; Baehr et al., 2004) and South Atlantic (e.g. Perez et al., 2011). The combined assimilation of open ocean hydrographic observations and the continuous RAPID array western boundary measurements have also been shown to locally improve the AMOC strength at 26.5°N (Stepanov et al., 2012). This emphasises the role that more systematic observations located at the eastern and western boundaries at several latitudes may play in monitoring the AMOC (Marotzke et al., 1999). In the future,

the SAMOC observing system (Ansorge et al., 2014; Hummels et al., 2015), which will provide time series of NBC measurements at the western boundary at 11°S, could be assimilated into the ORAs to constrain the regions of largest spread in the tropical South Atlantic.

Differences in data assimilation methods near the boundaries may also be influencing the overturning in the different ORAs. For example, Balmaseda et al. (2013) noted that the AMOC at 26°N in the ECMWF reanalyses is very sensitive to the treatment
of observations and the parametrization of their errors near to the boundaries, although similar changes are not documented for other ORAs. Stepanov et al. (2012) also showed that the assimilation impacts of the RAPID western boundary measurements on the AMOC can vary according to the prescribed horizontal scales of the DA error covariances, e.g. with boundary-focused covariances producing larger positive impacts on the AMOC than isotropic covariances. In order to better understand the large SAMOC sensitivity found between the ORAs, future work will focus on the response of the western
boundary and SAMOC transports to changes in the ORA configurations, such as sensitivity experiments to the assimilated datasets and to the DA schemes near to the western boundary.

Acknowledgments. The first author would like to acknowledge the financial support of the CAPES Foundation, Ministry of Education of Brazil (Proc. BEX 1386/15-8). The authors also would like to mention the support of the ORA providers and the

405 Copernicus Marine Service (http://marine.copernicus.eu/) to provide access of the reanalysis data used in this work.

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

# Tables

590

**Table 1.** List of the NEMO-based products used in this study and their central characteristics. Abbreviations: OSTIA stands for Operational Sea Surface Temperature and Sea Ice Analysis, AVISO for Archiving, Validation and Interpretation of Satellites Oceanography, AVHRR for Advanced Very High Resolution Radiometer, AMSR-E for Advanced Microwave Scanning Radiometer for Earth Observing System, NSIDC for National Snow and Ice Data Center, ICOADS for International Comprehensive Ocean-Atmosphere Data Set, NODC for National Oceanographic Data Center, EUMETSAT for European Organisation for the Exploitation of Meteorological Satellites, OSISAF for Ocean and Sea Ice Satellite Application Facility, CORA for Coriolis Dataset for Re-Analysis, and CERSAT for Centre ERS d'Archivage et de Traitement.

| Product   | Model /<br>Resolution                     | Atmospheric<br>Forcing                                   | Data assimilation                                                                                  | Assimilated observations                                                                                                                                                 | Initial conditions                                                                                   |
|-----------|-------------------------------------------|----------------------------------------------------------|----------------------------------------------------------------------------------------------------|--------------------------------------------------------------------------------------------------------------------------------------------------------------------------|------------------------------------------------------------------------------------------------------|
| ORCA025   | NEMO3.2 –<br>LIM2, 1/4°, 46<br>z-levels   | 6-hourly<br>ERA-Interim                                  | None                                                                                               | None                                                                                                                                                                     | 1/4° run with hydrographic data assimilation                                                         |
| ORCA0083  | NEMO3.2 –<br>LIM2, 1/12°,<br>75 z-levels  | 6-hourly DFS4.1<br>(1978 -2007) and<br>5.1 (2008-2010)   | None                                                                                               | None                                                                                                                                                                     | Levitus (1998) T/S climatology                                                                       |
| ORAP5     | NEMO3.4.1 –<br>LIM2, 1/4°, 75<br>z-levels | 6-hourly ERA-<br>Interim with wave<br>forcing            | NEMOVAR (3D-Var)<br>(Mogensen et al., 2012)                                                        | OSTIA SST, AVISO SLA, in situ T/S profiles<br>from EN3_v2 with bias correction for XBT,<br>OSTIA sea-ice concentration                                                   | 12-year spin-up initialised from<br>WOA09 T/S climatology and<br>followed by 5-year assimilation run |
| CGLORSV5  | NEMO3.2.1 –<br>LIM2, 1/4°, 50<br>z-levels | 3-hourly ERA-<br>Interim                                 | Global OceanVar (3D-Var)<br>(Storto et al., 2011)                                                  | Reynolds 1/4° AVHRR + AMSR-E SST,<br>AVISO SLA, in situ T/S profiles from EN3_v2<br>with bias correction for XBT, NSIDC ("NASA<br>Team" algorithm) sea ice concentration | Mean January condition of a 4-year<br>spin-up initialised from EN4 T/S<br>analysis                   |
| UR025.4   | NEMO3.2 –<br>LIM2, 1/4°, 75<br>z-levels   | 6-hourly ERA-<br>Interim                                 | Met Office FOAM – NEMO<br>assimilation system (Optimal<br>Interpolation)<br>(Storkey et al., 2010) | ICOADS in situ SST and NODC satellite SST,<br>AVISO SLA, in situ T/S profiles from EN3_v2<br>with bias correction for XBT, EUMETSAT<br>OSISAF sea ice concentration      | EN3 T/S analysis                                                                                     |
| GLORYS2V4 | NEMO3.1 –<br>LIM2, 1/4°, 75<br>z-levels   | 3-hourly ERA-<br>Interim with<br>RF and P<br>corrections | SAM2 (Singular Evolutive<br>Extended Kalman Filter)<br>(Pham et al., 1998)                         | Reynolds 1/4° AVHRR-only SST, AVISO SLA,<br>in situ T/S profiles from Coriolis CORA4.1<br>database, CERSAT sea ice concentration                                         | EN4 T/S analysis                                                                                     |

Figures

Figure 1. 3D schematic of the South Atlantic western boundary circulation and water masses from Soutelino et al. (2013). The water masses associated with the SAMOC upper limb are represented by the Tropical Water (TW), South Atlantic Central Water (SACW) and Antarctic Intermediate Water (AAIW). The circulation is represented by the Brazil Current (BC), Intermediate Western Boundary Current (IWBC), North Brazil Undercurrent (NBUC), North Brazil Current (NBC) and South Equatorial Current (SEC). The Deep Western Boundary Current (DWBC) is also shown flowing poleward along the NADW path.