# Peer review of "South Atlantic meridional transports from NEMO-based simulations and reanalyses"

_Ocean Science, 2017_

## Referee Comment (RC1) · Anonymous Referee #1 · 11 Oct 2017

The manuscript presents an analysis of differences of meridional heat and volume transports among different ocean simulations. Two free running models with with eddy resolving and eddy permitting resolution and four different data assimilation products have been investigated. The main finding it that the assimilation of data increases the transports, thereby bringing them into better agreement to independent estimates. The spread in heat transport is mainly related to volume transport differences which seem to be better constrained by the data in the interior that near the western boundary. The manuscript is rather descriptive and makes its point by many figures that have overlapping information content. The message does not go very deep but is well presented. Among the many different existing assimilation products the presented products are most similar yet differ in several aspects, which makes it difficult to figure out which

of the differences (if any) is key for different behavior. One wonders why not other products have been consulted or why no attempt has been made explore reasons for differences. However, there are no serious problems encountered and the manuscript may be publishable after minor revisions. Detailed comments are below.

**Details:**

L65: How about the contribution from the eastern boundary and the interior circulation, wouldn't these be worth to be shown or at least be mentioned?

L151: I am not clear which studies you refer to. There were already two named, and now two different follow. Maybe this could be slightly rearranged that it reads smoother.

**Fig. 4g: Label g missing**

L240: Presumably these are the same areas that contribute most to the MHT. The trivial expectation is that the relative spread is similar, such that differences in areas that matter most for the mean MHT also matter most for their spread. Is this so? Could you check this, maybe show the ensemble mean p-OTT.

L255-258: Wouldn't you expect to see an impact of the second peak of southward transports in the ORAs that the FRMs should not show? Also, since Fig.4 shows the mean, I don't see how you can infer conclusions for the time variability from this. You could investigate the contributions to the heat transport variability in more detail instead of speculating.

L272-273: I think showing the ensemble mean p-OTT would also serve here to make this point. Fig.1 shows the volume transport but the depth integrated heat transport could be different.

L278-279: It would be nice to add information on these limits to the figure caption or state them somewhere else.

L294-296: Isn't this basically what we already know from Figure 2?

L313-314: Can this variability be considered realistic? Are the associated features similar to the high resolution model simulation? For instance, the ORAs, except for ORA-IP, have substantially more variability in the interior than the eddy resolving model.

L323-325: It does not become clear why these two time series are shown together. What is their relation or the intention here?

СЗ

---

## Author Comment (AC1) · 29 Nov 2017

The response to the reviewer is in the pdf file which can be accessed with the link below

Please also note the supplement to this comment:
https://www.ocean-sci-discuss.net/os-2017-69/os-2017-69-AC1-supplement.pdf

———————————————

---

## Author Response (AR1)

**RESPONSE TO REVIEWER 1**

The authors would like to thank Reviewer 1 for the comments and suggestions that have helped to improve the overall quality of the paper. In our answers, we have referred to the line numbers as in the marked-up manuscript version.

**General comments:**

**One wonders why not other products have been consulted or why no attempt has been made to explore reasons for differences.**

The NEMO-based Ocean Reanalyses (ORAs) used in this work, which are quite similar products among many different existing ORAs (as pointed by the reviewer), already show large differences in the South Atlantic meridional transports. Including more and different reanalyses would only add an extra factor – different models – to be accounted for in explaining differences. We also wanted to focus on a comparison between ORAs and Free-Running Models (FRMs) and to explore differences/similarities between them, so we use FRMs and ORAs with the same basic NEMO model. We think this dataset (totalling 6 products) allows to compare the ORAs and FRMs, as well as to explore the ORA transport differences in the South Atlantic.

We agree with the reviewer that more work is needed to properly investigate the reasons for the ORA differences, but we have considerably narrowed the problem to understand how to better constrain the western boundary transports. We have shown that the current ORAs do show good agreement in currents and transports in the interior which is a big step forward. We make clear in the conclusions that future work will address specifically this western boundary problem. We intend to use one specific ORA product and run sensitivity experiments changing the data assimilation configurations near the western boundary so that we have control over the product to better address the western boundary issue. Also because we do not have a good observational truth for these western boundary flows, in our opinion this needs a different approach, rather than intercomparison of current products, to continue the investigation. These next steps to fully pin down the reasons behind the inter-product spread represent a complete study in itself.

**Details:**

**L72: How about the contribution from the eastern boundary and the interior circulation, wouldn't these be worth to be shown or at least be mentioned?**

We agree with the reviewer about mentioning the interior and eastern boundary. Both are now mentioned in the sentence starting in L70, as below:

"Their result reveals the need for further assessment of the skills and uncertainties of the ORAs in the South Atlantic, such as comparing them with Free-Running Models (FRMs) and evaluating their SAMOC contributions across the eastern, interior, and western boundary regions shown in Fig. 1."

However, for Fig.1 we still focus on the western boundary circulation to set the scene, particularly for the analyses of Fig. 9, Fig. 10 and Fig. 11 in the section 3.3 showing the western boundary role in the large-scale transports.

**L163: I am not clear which studies you refer to. There were already two named, and now two different follow. Maybe this be slightly rearranged that it reads smoother.**

The sentence in L161 was rearranged as below:

"The large-scale transports are compared to the 34 high-density XBT-based estimates (XBT-AX18) in the Southern Atlantic from 2002 to 2013, with transport estimates at 35˚S and 30˚S given by Majumder et al. (2016). Recent observational studies are also used for comparison, which employ different methodologies to calculate the SAMOC and MHT between 35˚S and 20˚S, as follows: (i) an Argo climatology (Dong et al., 2014), (ii) altimetry synthetic profiles based on the correlation of the AVISO SLA and isotherm depths (Dong et al., 2015), and (iii) dynamic height fields from Argo and AVISO SSH (Majumder et al., 2016) are used together with wind fields to estimate the total transports."

**Fig. 4g: Label g missing**

Label g is now included in Fig. 4.

**L258: Presumably these are the same areas that contribute most to the MHT. The trivial expectation is that the relative spread is similar, such that differences in areas that matter most for the mean MHT also matter most for their spread. Is this so? Could you check this, maybe show the ensemble mean p-OTT.**

According to the plot below, areas of largest spread do correspond to areas with largest mean transports (i.e. along the western boundary).

[Figure]

We now note it in the text, which has been changed in L257 as below:

"The $v\bar{T}$ component captures variations from ~0.2 PW to 1 PW (Fig. 7c), explaining ~83% of the total MHT spread which is mainly concentrated in the areas with largest mean transports, i.e. the narrow western boundary region (Fig. 7d)."

**L274-278: Wouldn't you expect to see an impact of the second peak of southward transports in the ORAs that the FRMs should not show? Also, since Fig.4 shows the mean, I don't see how you can infer conclusions for the time variability from this. You could investigate the contributions to the heat transport variability in more detail instead of speculating.**

The ORAs southward peak between 10°S and 5°S in the maps of Fig. 4 does actually increase their southward flow by ~4 Sv compared with the FRMs (see Fig. 4g). However their North Brazil Current (NBC) transports can increase from ~4 up to 9 Sv compared to the FRMs (see Fig. 5). So, the ORA second southward peak is more than balanced by the increase in the western boundary transports, and this is why there are no clear sings of it in Fig. 2a or in Fig. 3c.

Answering the second part of the reviewer's question, the interior southward flow increases towards the equator for all products, reaching similar magnitudes to the overturning component, and with both having similar $\Delta T$s. So it is natural to expect that variations in the southward flow will also contribute to the MHT variability in this region. This is all that was intended. Further investigation of this time variability would be possible but is not the main focus of this paper. The sentence is now modified in L276 as below:

"Therefore it is likely that these large upper level tropical circulations explain why $\psi_{max}$ does not dominate the MHT variability close to the equator, as also noted by Valdivieso et al. (2014). "

**L293-294: I think showing the ensemble mean p-OTT would also serve here to make this point. Fig. 1 shows the volume transport but the depth integrated heat transport could be different.**

We have added this figure in answer to L258 above but we do not see how it is relevant to this point? The main point of Fig. 9 is to show the continuous band of positive MHT regression coefficients against the western boundary which means that ORAs with the largest MHTs must show less southward transport along the Brazil Current (BC) and higher northward transport along the NBC, compared to the FRMs. This pattern is further confirmed by Fig. 10 for both BC and NBC.

**L299-300: It would be nice to add information on these limits to the figure caption or state them somewhere else.**

We agree with the reviewer. The water masses limits are now stated in the captions of Fig. 10:

"The TW, SACW and AAIW limits are defined in kg m$^{-3}$ with σ < 25.5, 25.5 ≤ σ < 27.1, and 27.1 ≤ σ < 27.3, respectively."

**L315-317: Isn't this basically what we already know from Figure 2?**

In Fig. 2 we show the AMOC strength across the basin which is a zonally integrated quantity. In the lines here we discuss the 4-box model of the transports (Fig. 11), i.e. the transports are broken down into western boundary versus interior, and into upper versus deep circulations. We agree that main difference between the products is shown in Fig. 2 but it cannot be really understood. Fig. 11 shows where the inter-product compensations occur and how the 4-box transports are balanced within products. We therefore say that GLORYS2V4 and UR025.4 are ~10 Sv and 8.5 Sv larger than ORCA025 in the upper western boundary, and these inter-product differences are mostly compensated by the deep western boundary transports. In Fig. 11, we can also see the contributions and their much better agreement from the interior boxes.

**L334-335: Can this variability be considered realistic? Are the associated features similar to the high resolution model simulation? For instance, the ORAs, except ORAP5, have substantially more variability in the interior than the eddy resolving model.**

Masina et al. (2015) also found that these NEMO reanalyses show an increased Eddy Kinetic Energy (EKE) and that they are in much better agreement with the OSCAR estimates when compared to no-assimilation runs. GLORYS, CGLORS and UR025-4, which have the largest transport variability here, also show the best level of agreement with OSCAR. This is an indication that the higher ORA variability caused by DA in the velocity fields is consistent with information inferred from observations. We have tried to be more specific changing the text in L334 as below:

"Figures 12a-f show that the interannual variability in p-OTTs is larger in the ORAs and in the high resolution ORCA0083 than in ORCA025. The assimilation of observations in eddy-permitting models introduces variability that would otherwise only appear with higher resolution, as in ORCA0083. According to Masina et al. (2015), this higher variability in the ORAs is in better agreement with the Eddy Kinetic Energy estimates from the ocean surface current velocities (OSCAR) product than that of the FRMs."

**L353-355: It does not become clear why these two time series are shown together. What is their relation or the intention here?**

The intention is to verify how these two components of the circulation behave over time in the tropical South Atlantic. For example, how are these transports in the ORAs impacted by the introduction of Argo? From Fig. 13 the southward interior flow is better constrained by the ORAs in the later years, related with the assimilation of a larger number of (Argo) hydrographic observations (including salinity observations) across the basin. However, the overturning component is dominated by the narrow western boundary and the lack of observations in these narrow areas means the overturning is not better constrained, and so the ENS-ORA spread of the overturning component remains nearly steady over time.

**RESPONSE TO REVIEWER 2**

The authors would like to thank Reviewer 2 for the comments and suggestions that have helped to improve the overall quality of the paper. In our answers, we have referred to the line numbers as in the marked-up manuscript version.

**General comments:**

**Much of it is either unsurprising (ORAs are closer to observations than FRMs) or under-explored (what needs to be done in the ORAs to better represent the boundary currents).**

We provide a comprehensive analysis showing where the differences between these products come from (i.e. the western boundary currents). The results make sense in relation to the data that have been assimilated. Although the results are perhaps not surprising, they had not be shown with this level of clarity in previous studies. In addition, the consistent impact of assimilation on the interior circulations in multiple products has not been previously shown. The comparison with the FRMs clearly shows that ORAs have additional circulation differences relative to the FRMs in the basin interior.

We agree with the reviewer that there is more work to be done in order to fully understand the reasons for the ORA differences at the western boundary. However, this requires a full study in itself. A different approach making sensitivity experiments in the South Atlantic with a single ORA, changing for example data assimilation configurations and assimilated observations near the western boundary, is needed to make further progress. This is currently being done and will hopefully come out as the future paper.

**Just from reading the manuscript, it is not clear how this study is different from the Majumder et al. (2016) paper. This needs to be better highlighted in the introduction section.**

Majumder et al. (2016) have used ORAs and observational estimates to study the meridional transports in the Southern Atlantic. However, their study does not go further than showing that the transport magnitudes between ORAs as well as between ORAs and observations show large discrepancies. Motivated by Majudmer et al. (2016), we show here more details of the ORA transport differences and interpret the impact and limitations of the DA schemes in improving the South Atlantic circulation, also through the inclusion of FRMs with distinct spatial resolutions. For example, we clearly demonstrate the dominant role of inter-product spread in velocity, and the limited contribution of inter-model spread in temperature. This is now better highlighted in the text - L71 and L83 - as below:

L70: "Their result reveals the need for further assessment of the skills and uncertainties of the ORAs in the South Atlantic, such as comparing them with Free-Running Models (FRMs) and evaluating their SAMOC contributions across the eastern, interior, and western boundary regions shown in Fig. 1."

L82: "Going further than Majumder et al. (2016), we also narrow down these transport differences in an attempt to understand the potential impact (and limitations) of the DA schemes in improving the ORA states in the South Atlantic."

**It would be good to discuss the findings, especially that the models so poorly represent the boundary currents, in relation to OSSEs; i.e. which measurements are needed (and where are they needed) to improve the ORAs?**

We have now included more discussion about model sensitivity studies and possible OSSEs. The text has been changed in L400 as below:

"Observation system simulation experiments (OSSEs) with AMOC trans-basin arrays have shown that the meridional flow strength can be sensitive to the number of hydrographic profiles near the boundaries in both North (e.g. Hirschi et al., 2003; Baehr et al., 2004) and South Atlantic (e.g. Perez et al., 2011). The combined assimilation of open ocean hydrographic observations and the continuous RAPID array western boundary measurements have also been shown to locally improve the AMOC strength at 26.5˚N (Stepanov et al., 2012). This emphasises the role that more systematic observations located at the eastern and western boundaries at several latitudes may play in monitoring the AMOC (Marotzke et al., 1999). In the future, the SAMOC observing system (Ansorge et al., 2014; Hummels et al., 2015), which will provide time series of NBC measurements at the western boundary at 11˚S, could be assimilated into the ORAs to constrain the regions of largest spread in the tropical South Atlantic."

**I was surprised to read that the ORAs are much less constrained before the 2000s, yet the authors chose to use the 1997-2010 period for their analysis. Why not start after the Argo era begins then? And why not include later years; 2010 is seven years ago! That would make much more sense to me.**

The UR025.4 reanalysis ends in 2010, so we took 2010 as the final year of the chosen period. In addition, the plot below show that meridional transport differences for the periods 1997-2010 (solid lines) and 2000-2010 (dashed lines) are quite small.

[Figure]

Including some years before the Argo period also allows us to show that the ORA interior circulation is better constrained after the Argo begins (see Fig. 13), whereas the overturning spread between them remains steady over time. If we had only used Argo period results we would have had few averaging years. We have modified the paragraph in L153 as below:

"In order to avoid any dynamical spin-up in the early years of the simulation for products starting in the late eighties or early nineties (e.g. UR025.4 and GLORYS2V4), and because UR025.4 ends in 2010, a common time period from 1997 to 2010 is chosen. Despite the fact that subsurface ocean observations are scarcer before the 2000s (i.e. prior to the full deployment of Argo floats), the total meridional transports for the periods 1997-2010 and 2000-2010 do not differ significantly. "

**The authors need to state much more clearly which of the observational data they use to assess the skill of the models have gone into the ORAs themselves, i.e. is the XBT-AX18 line, or WOA13, used in the data assimilation process of the ORAs? In other words, are these independent validations?**

We agree with the reviewer about this comment and this is now clearly mentioned in a new paragraph starting in L174:

"Of the observational estimates above, the XBT-AX18 line is not independent as it is included in the EN3 and EN4 datasets which are assimilated by the ORAs (see Tab.1). Although WOA13 is not directly assimilated by the ORAs, it uses the same observational information as EN3 and EN4, and so it also cannot be treated as completely independent."

**There could be more discussion of the features seen in the Cape Basin. Where do these come from? How are they related to MOC and the Brazil Current? Are they noise or signal?**

From checking spatial animations of SLA in the Cape basin as well as maps of Eddy Kinetic Energy for the products, it is clear that there are differences in reproducing the Agulhas rings between the FRMs and ORAs, as well as between the ORAs. This implies that the Cape basin variability impacts the ENS-ORA and ENS-ALL transport spreads (Fig. 9). This may be related to the way SLA is assimilated into the ORAs, see added discussion:

L288: "South of 25°S the p-OTT contributions to the total MHT are more distributed, with a noticeable contribution from the Agulhas leakage caused by the different intensity and positioning of the Agulhas rings between the products as they travel westward across the Cape basin."

L346: "The different levels of variability in the Agulhas leakage between ORCA025 and ORAs may be attributed to the impacts of the SLA assimilation (Backeberg et al., 2014). However, even between ORAs these Agulhas patterns differ, e.g. the weaker contributions in ORAP5 may be due to smoothing from the super-observation method applied to the altimeter data (Mogensen et al., 2012), as also noted by Masina et al. (2015)."

**For a study that focusses so much on boundary currents, it is surprising that there is no mention of how the boundary conditions have been implemented in the models? Are these all the same? Partial slip? What parameters have been used? Is there any relation between the way the boundary conditions are implemented and the skill of the ORAs/FRMs?**

We agree with this comment and information about the lateral boundary conditions for all the products are now given in the text:

L121: "Both ORCA0083 and ORCA025 employ a free-slip (no-stress) configuration for the lateral momentum boundary conditions."

L148: "On lateral boundaries, UR025.4 and ORAP5 adopt a free-slip configuration whereas CGLORSV5 and GLORYS2V4 employ a partial-slip condition. In the latter, the constant of proportionality ($\alpha$) between the tangential stress and the tangential velocity is defined as 0.5 for both products."

Some further discussion has been added in L394 with respect to the lateral boundary conditions:

"It is noteworthy that the lateral boundary conditions in the ORAs and FRMs vary between free-slip ($\alpha=0$) and partial-slip ($\alpha=0.5$). However, there is no clear correspondence between the choice of lateral boundary conditions and the strength of the western boundary transports, with free-slip products (e.g. UR025.4) having similar transports to partial-slip products (e.g. GLORYS2V4)."

**Minor issues:**

**- line 14: Start the abstract with a sentence about why the South Atlantic is relevant? The introduction section starts with a few good paragraphs, and these might be summarised at the beginning of the abstract?**

The abstract now starts with the following sentence:

"The Meridional Heat Transport (MHT) of the South Atlantic plays a key role in the global heat budget: it is the only equatorward basin-scale ocean heat transport and it sets the northward direction of the global cross-equatorial transport."

We also thought we could better improve the abstract as a whole, making it easier to understand for a broader number of readers. So you will find a few changes and text rearrangements there.

**- line 49: Add 'possibly' before 'leading' ?**

Done.

**- line 65: Explicitly state that most ORAs lack dynamical consistency?**

We have removed the word "dynamically" from the ORA definition in L64. We think this is enough to meet the reviewer's requirement.

**- line 113: I thought most NEMO models were z*?**

For all the NEMO-based products used here, the code versions with fixed z-levels and partial cell topography are used. There are additional challenges in implementing DA (especially altimetry) with z*.

**- line 119: What do 'W', 'SAT' and 'SAH' stand for?**

These acronyms are defined in L106 and L107:

"The ERA-Interim reanalysis provides Winds (W) at 10 m, Surface Air Temperature (SAT) and Surface Air Humidity (SAH) at 2 m, daily Radiative Fluxes (RF) and Precipitation (P) fields…"

**- line 263: Why is this interpretation made? What is the motivation for this?**

There is a clear line of spread in the p-$\bar{v}T$ right against the western boundary (Fig. 7f). This might indicate that local response to small temperature changes on the western boundary slope may largely determine the meridional transport variability in ocean models through changes in the western boundary current velocities (p-$v\bar{T}$; Fig. 7d), as already stated by Bingham and Hughes (2009) for the North Atlantic (as discussed in the paper in L390).

**- Figure 2: How is 'spread' defined here? Highest-lowest? Standard deviation?**

The spread is defined as the standard deviation between the products. To make this point clear, we have added this information to the caption of Fig. 2:

"(a) The AMOC strength $\psi_{max}$ (Sv) averaged over 1997-2010 as a function of latitude, and (b) its spread (Sv) defined as the standard deviation of the ENS-ALL and ENS-ORA."

**- Figure 3: Why does the size of the horizontal lines on the whiskers vary? What does this mean?**

The horizontal lines on the whiskers do not mean anything relevant. However, the size of these horizontal lines in Fig. 3c were adjusted to be the same size.

**- Figure 5: How is the standard error defined here?**

It is defined as the standard deviation divided by the square root of the length of the monthly time series. The caption of Fig. 5 is now changed to include the following information:

"The black bars represent the standard errors where the size of the sample is defined as the length of the monthly time series."

**- Figure 8: What is the relevance at this line of 0.7 correlation? Is this significance level? How calculated?**

We have removed the 0.7 line from the figure to avoid any kind of misunderstanding. In Fig. 8, we compute Pearson correlations calculated with 95% significance level on a monthly time scale. This information is now added to the caption of Fig. 8:

"The monthly Pearson correlation between the SAMOC strength and the MHT as a function of latitude for 1997-2010, calculated with significance level of 95%. "

**- Figure 11: What do the dashed circles represent above the top two bar charts?**

It was an attempt to say that in the tropical South Atlantic the northward upper western boundary has part of its transports to compensate for the southward deep western boundary flow (solid circles), and other part to compensate for the southward upper interior flow (dashed circles). However, we have removed the dashed circles in order to avoid any kind of misunderstanding.

**- line 29: The word 'right' is confusing here, as it might also be interpreted as the right (as opposed to left) side of the plots?**

We have removed the entire sentence of L26 since we have changed the abstract as a whole.

**- line 81: 'Focusing on' and line 412: 'near the boundaries'**

Done.

**- line 267: Is this correlation R or R^2?**

R.

[revised manuscript text omitted]

**Figures**

[Figure]

**Figure 1.** 3D schematic of the South Atlantic western boundary circulation and water masses from Soutelino et al. (2013). The water masses associated with the SAMOC upper limb are represented by the Tropical Water (TW), South Atlantic Central Water (SACW) and Antarctic Intermediate Water (AAIW). The circulation is represented by the Brazil Current (BC), Intermediate Western Boundary Current (IWBC), North Brazil Undercurrent (NBUC), North Brazil Current (NBC) and South Equatorial Current (SEC). The Deep Western Boundary Current (DWBC) is also shown flowing poleward along the NADW path.

[Figure]

**Figure 2.** (a) The AMOC strength $\psi_{max}$ (Sv) averaged over 1997-2010 as a function of latitude, and (b) its spread (Sv) defined as the standard deviation of for the ENS-ALL and ENS-ORA. The black box represents the study area between 35˚S and the equator.

[Figure]

**Figure 3.** (a) SAMOC strength (Sv) at 35˚S, (b) MHT (PW) at 35˚S, and (c) MHT (PW) as a function of latitude averaged over 1997-2010. The black bars in (a) and (b) represent monthly standard deviations, except for the XBT-AX18, Dong et al. (2015) and Majumder et al. (2016) estimates which correspond to quarterly, weekly and daily standard deviations, respectively. In (c), Liu et al. (2015)'s MHTs and their annual standard deviation are represented by the shaded grey area. The products are also compared to hydrographic and inverse modelling estimates from the literature at several latitudes.

[Figure]

640

**Figure 4.** East-west accumulated volume transports (1997-2010) for each product (a to f) calculated from the surface down to $z_{max}$ at each latitude. The upper southward flow in (g) is defined by the southward maximum of the east-west accumulated volume transports. Units are in Sv and the black dashed contour corresponds to 0 Sv.

645

[Figure]

**Figure 5.** The NBC transports (1997-2010) at 11˚S calculated between the surface and the neutral density interface of 27.7 kg m$^{-3}$, using the same section near the western boundary and methodology as in Hummels et al. (2015). The black bars represent the standard errors where the size of the sample is defined as the length of the monthly time series. and the Uunits are in Sv.

650

655

[Figure]

**Figure 6.** The zonal averaged temperature (˚C) as a function of latitude for WOA13 from 1995 to 2012 (top panel), followed by the zonal averaged temperature of each product from 1997-2010 minus WOA13. The  thick solid line represents the  0˚C contour.

[Figure]

**Figure 7.**  (Left) The original MHTs (a), the MHTs based on the $v\bar{T}$ component (c), and the MHTs based on the $\bar{v}T$ component (e) in PW.  (Right) The ENS-ALL spread of the p-OTTs (b), p-$v\bar{T}$ (d) and p-$\bar{v}T$ (f)  in PWT per 0.25°. Overbar represents the mean of the ENS-ALL.

[Figure]

**Figure 8.** The monthly Pearson correlation between the SAMOC strength and the MHT as a function of latitude for 1997-2010, calculated with significance level of 95%. The quarterly XBT-AX18 correlation between the SAMOC strength and MHT at 35˚S is also included for comparisons. The black solid line represents the correlation of 0.7.

670

675

680

[Figure]

**Figure 9.** The linear regression coefficient between the inter-product p-OTTs and their MHTs for each latitude. Units are in PWT per 0.25˚ per 1 PW across each latitude.

685

[Figure]

**Figure 10.** The transports (Sv) within 6˚ of the west coast for the (a) BC and (b) IWBC-NBUC-NBC system, following the isopycnal limits of the South Atlantic western boundary water masses as in Mémery et al. (2000) and Donners et al. (2005). The TW, SACW and AAIW limits are defined in kg m$^{-3}$ with σ < 25.5, 25.5 ≤ σ < 27.1, and 27.1 ≤ σ < 27.3, respectively. Their ENS-ORA spreads (Sv) of the western boundary current transports (Sv) are displayed in (c).

[Figure]

**Figure 11.** 4-box model of the averaged transports (1997-2010, in Sv) from (a) 15˚S to the equator, and from (b) 30˚S to 15˚S. 6˚ off the coast is chosen to separate the western boundary  from the basin interior The depth of maximum SAMOC $z_{max}$ for each product is used to separate the upper and deep circulations. The circles with "x" and dots represent flow going into and out of the page, respectively. The empty circle means that there is no agreement about the direction of the flow. ± corresponds to the interannual variability of each product

[Figure]

**Figure 12.** (a-f) Interannual p-OTT spread for the period 1997-2010. Units are in PWT per 0.25˚. In (g) the interannual p-OTTs variances for each product are summed within 6˚ of the west coast across each latitude and displayed as a percentage of the total MHT variance.

[Figure]

715

**Figure 13.** (a) Monthly time series of $\psi_{max}$ (Sv) and the maximum upper southward flow (Sv) for each product calculated as an average from 15˚S to the equator, and. In (b) theirtheir ENS-ORA spreads (Sv) are presented. A running mean of 6 months was applied to smooth the ENS-ORA spread time series. The upper southward flow is calculated using the same approach as in Fig. 4g.